# Research priorities for children's cancer: a James Lind Alliance Priority Setting Partnership in the UK

Susie Aldiss [1], Rachel Hollis,[2] Bob Phillips,[3,4] Ashley Ball-Gamble [5] Alex Brownsdon,[6] Julia Chisholm,[7,8] Scott Crowther,[9] Rachel Dommett,[10] Jonathan Gower,[11] Nigel J Hall [12,13] Helen Hartley,[14] Jenni Hatton,[15] Louise Henry,[7] Loveday Langton,[16] Kirsty Maddock,[2] Sonia Malik,[17] Keeley McEvoy,[18] Jessica Elizabeth Morgan [3,19] Helen Morris,[20] Simon Parke,[21] Sue Picton,[2] Rosa Reed-Berendt,[22] Dan Saunders,[23] Andy Stewart,[24] Wendy Tarplee-Morris,[25] Amy Walsh,[26] Anna Watkins,[16] David Weller,[27] Faith Gibson [1,28]

For numbered affiliations see end of article.

**Correspondence to**
Susie Aldiss;
s.aldiss@surrey.ac.uk

## ABSTRACT

**Objectives** To engage children who have experienced cancer, childhood cancer survivors, their families and professionals to systematically identify and prioritise research questions about childhood cancer to inform the future research agenda.

**Design** James Lind Alliance Priority Setting Partnership.

**Setting** UK health service and community.

**Methods** A steering group oversaw the initiative. Potential research questions were collected in an online survey, then checked to ensure they were unanswered. Shortlisting via a second online survey identified the highest priority questions. A parallel process with children was undertaken. A final consensus workshop was held to determine the Top 10 priorities.

**Participants** Children and survivors of childhood cancer, diagnosed before age 16, their families, friends and professionals who work with this population.

**Results** Four hundred and eighty-eight people submitted 1299 potential questions. These were refined into 108 unique questions; 4 were already answered and 3 were under active study, therefore, removed. Three hundred and twenty-seven respondents completed the shortlisting survey. Seventy-one children submitted questions in the children's surveys, eight children attended a workshop to prioritise these questions. The Top 5 questions from children were taken to the final workshop where 23 questions in total were discussed by 25 participants (young adults, carers and professionals). The top priority was 'can we find effective and kinder (less burdensome, more tolerable, with fewer short and long-term effects) treatments for children with cancer, including relapsed cancer?'

**Conclusions** We have identified research priorities for children's cancer from the perspectives of children, survivors, their families and the professionals who care for them. Questions reflect the breadth of the cancer experience, including diagnosis, relapse, hospital experience, support during/after treatment and the long-term impact of cancer. These should inform funding of future research as they are the questions that matter most to the people who could benefit from research.

## STRENGTHS AND LIMITATIONS OF THIS STUDY

⇒ We made use of the well-established and transparent James Lind Alliance methodology and clearly describe the process and decision-making which led to the final Top 10 research priorities.

⇒ The process followed ensures that these priorities came directly from those who are the most affected by childhood cancer but rarely influence the research agenda.

⇒ We ensured the priorities of patients/survivors, parents/relatives/friends and professionals were given equal weighting at the interim priority setting stage.

⇒ We used innovative methods to hear directly from children about their priorities for future research through surveys and a workshop specifically designed for them.

⇒ Under-represented groups in the survey submissions included people from minority ethnic groups, males and primary healthcare professionals.

## INTRODUCTION

Annually, there are around 1800 new cases of cancer in children in the UK.[1] While research over the last four decades has dramatically increased the overall 5-year survival rate for all childhood cancers to around 84%,[2] further research is needed to not only improve outcomes for all types of cancer, but to support all children to live long, healthy and happy lives.

Historically, topics of healthcare research in children's cancer have been driven by perspectives of researchers and the pharmaceutical industry, meaning what is most important to children, survivors, their families and the professionals who care for them, has sometimes been overlooked. Prioritising areas for research as identified by children

and carers is crucial. There is increasing evidence that research questions and outcomes prioritised by professionals may not be aligned to those experiencing the disease.[3] Patients and carers tend to prioritise non-drug treatment research while ongoing research strategies are dominated by drug evaluations.[4] This mismatch in priorities is particularly relevant for children due to their unique physiological and psychosocial status and relative rarity of cancer. Increasingly, research funders are asking if proposed research is a priority for patients.

The James Lind Alliance (JLA) is a non-profit-making initiative bringing together patients, carers and professionals in Priority Setting Partnerships (PSPs) focusing on specific health conditions http://www.jla.nihr.ac. uk/priority-setting-partnerships/). JLA PSPs identify and prioritise unanswered questions, so researchers and research funders are aware of the issues that matter most to those who could benefit from that research.[5]

In 2019, Children's Cancer and Leukaemia Group (https://www.cclg.org.uk/) and The Little Princess Trust (https://www.littleprincesses.org.uk/) partnered with the JLA on the Children's Cancer PSP. One of our primary goals was to prioritise the voice of children about what research should be undertaken. Previous PSPs have sought to involve children and young people, but in the final reporting it is evident that few children, especially young children, had been engaged through the process.[6] We recognised the challenges of engaging with these populations, in terms of reach and accessibility of information and determined we would invest time and resources, in exploring and resolving any challenges that could impact on participation.

Following the JLA methodology, we aimed to conduct a UK-wide research prioritisation exercise for childhood cancer to inform decisions of research funders and support the case for research in this underserved group.[7]

## METHODS
Methodology followed the JLA process,[5] the protocol is available from: https://www.jla.nihr.ac.uk/priority-setting-partnerships/childrens-cancer/.

## Setup
### Project management
There was a coordinating team of four researchers, nurses and clinicians. An expert steering group (all coauthors) oversaw the project, approved aims/objectives, survey materials, contributed to data analysis and summary question formation, and provided expert opinions for evidence checking. The steering group included parents of a child with cancer (n=5); an adult survivor of childhood cancer; a range of professionals reflecting the multidisciplinary nature of the care of children with cancer including: a teacher, general practitioner, surgeon, pharmacist, dietitian, speech and language therapist, clinical psychologist, physiotherapist, nurses (n=2), doctors (n=6) and representatives from the third sector (n=3), including the charities funding the project. The JLA chair (JG) provided neutral facilitation of meetings. The steering group identified potential partners, mainly children's cancer charities and professional networks, who were approached to assist with survey dissemination.

### Scope
This project focused on cancer and cancer-like conditions in children aged 0 to <16 at initial diagnosis. The scope, kept intentionally broad, included questions on any aspect of the cancer experience (figure 1).

Our aim was 'to identify gaps and unanswered questions in research about children's cancer from patients, carers and professionals' perspectives and then prioritise those that these groups agree are the most important for research to address.'

### Process
Figure 2 summarises the complete process.

### Stage 1a: gathering questions: initial survey
The survey was developed by the steering group and built using Qualtrics software. It was piloted with eight adult survivors of childhood cancer, nine parents and two professionals outside the steering group and adapted to incorporate their feedback. The survey launched on 9 September 2020 and closed on 8 January 2021. The following groups were invited to participate:

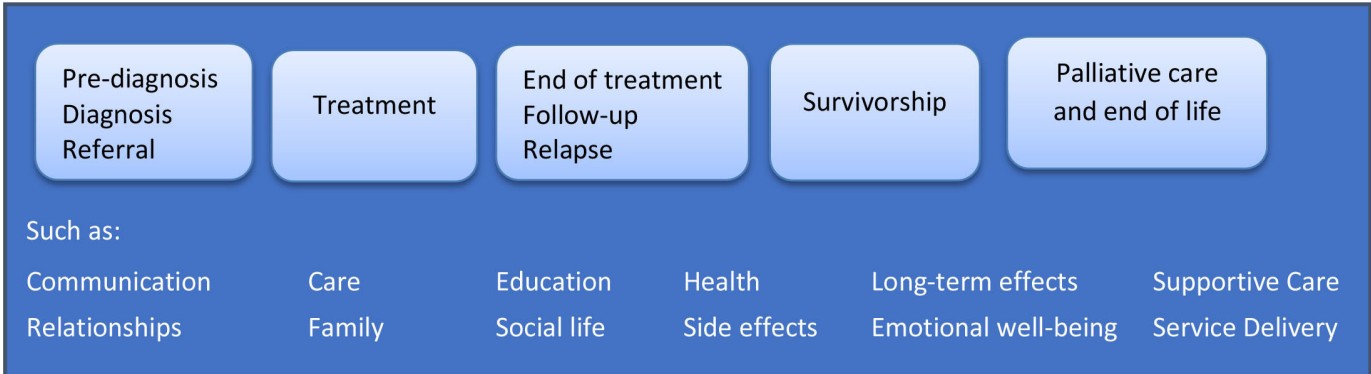

**Figure 1** Pathway of care included in the project scope.

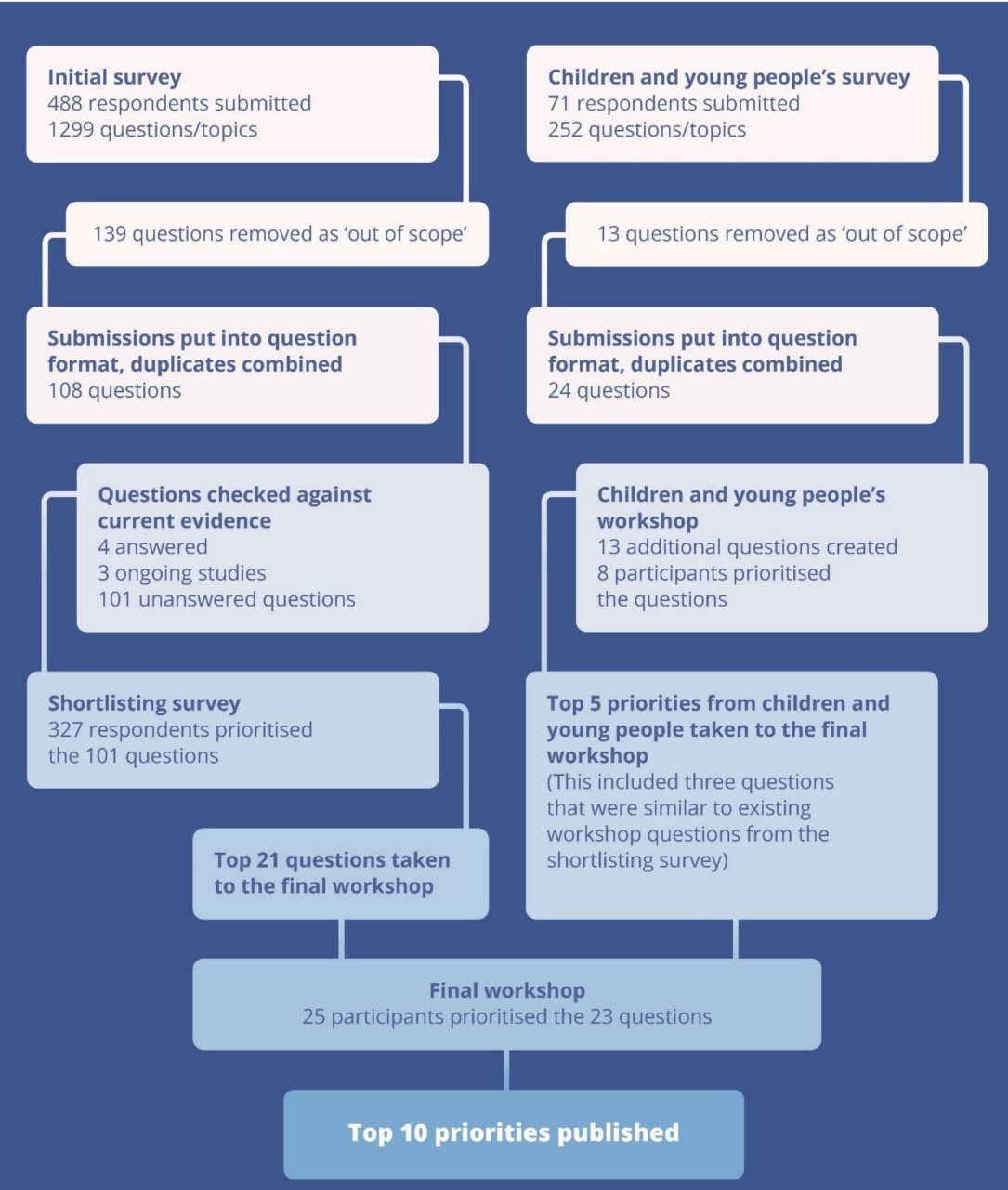

**Figure 2** Overview of the Children's Cancer Priority Setting Partnership methodology and results.

► People diagnosed with cancer before their 16th birthday.
► Relatives/friends/partners/carers of someone diagnosed with cancer before their 16th birthday.
► Professionals involved in diagnosing or treating children who have cancer or had cancer under 16.
► Professionals involved in the care of children who have cancer or had cancer under 16 and/or their families.

Respondents could submit up to eight questions about any aspect of children's cancer they considered important and unanswered. Basic demographic data were requested, and a box was available for free-text comments. Partners promoted the survey through websites, social media, newsletters and email.

**Stage 1b: gathering questions from children and young people**

A subgroup of the steering group was established to focus on our engagement with children. This consisted of two researchers, a teacher, doctor, health play specialist, parent, clinical psychologist and charity representative. Our initial intention had been to run a series of face-to-face workshops with children to collect questions, this was not possible due to the COVID-19 pandemic until the final workshop in the PSP process.

We determined that the best way to reach children would be through their parents/carers. Three survey versions were built using Qualtrics software, aimed at children of different ages (4–7, 8–12, 13–15 years). Surveys were piloted with three children and young people. They

varied in complexity of language used in the introduction and questions, and surveys for young people contained more questions seeking demographic information: participants could complete whichever survey version they preferred. Animations were developed to assist parents explain the project and survey to their child(ren) (surveys/animations available here:https://www.jla.nihr.ac.uk/priority-setting-partnerships/childrens-cancer/).

Surveys were launched on 6 September 2021 and closed on 16 November 2021 inviting participants who:

► Were diagnosed with cancer before their 16th birthday.
► Have a brother or sister with cancer now or who had cancer when they were younger (diagnosed before their 16th birthday).
► Have a friend with cancer now or who had cancer when they were younger (diagnosed before their 16th birthday).

Respondents were invited to submit up to eight questions/topics about any aspect of children's cancer they considered important. Surveys were promoted through the PSP's Partners, social media and posters were sent to all UK Principal Treatment Centres.

### Stage 2a: refining questions from the initial survey
Submitted questions were examined in detail and free-text sections studied for further questions.

#### Organising the questions
Initial coding was carried out by coordinating team members (SA and FG). Questions were grouped into themes. During coding, potential 'out-of-scope' questions were identified (see box 1 for criteria used). Identification of out-of-scope questions was an iterative process, checked and agreed by the steering group.

Similar questions were grouped to form summary questions. The aim was to retain the sense of what respondents meant, but in the form of a clear question. Steering group members met online in small groups to review summary questions within their area of expertise/experience, to confirm the grouping of questions, and wording

of each summary question. The steering group reviewed the whole summary question list.

#### Evidence searching
Searches were undertaken to identify questions answered by existing evidence. A search strategy was agreed with the steering group (see question verification form:https://www.jla.nihr.ac.uk/priority-setting-partnerships/childrens-cancer/). Searches were carried out by SA in January–May 2022. Searches were limited to evidence published in the last 5 years (since January 2017) and focused on evidence gathered from multiple studies (eg, systematic reviews, qualitative meta-synthesis). Searches were undertaken for ongoing studies which included personal communication with experts in the field and steering group members' knowledge of current research.

### Stage 2b: refining questions from children and young people
The same process was followed for refining questions from the children and young people's surveys. Questions were grouped into themes by SA with support from FG, similar questions were merged, and summary questions created. Out-of-scope questions were removed, if they were unrelated to cancer or were unclear (eg, 'cost to hospital', 'wildlife'). The subgroup met online to review summary questions and out-of-scope questions, with further checking undertaken via email until agreement was reached.

### Stage 3: question prioritisation
#### Shortlisting survey preparation
The steering group discussed whether to take all unanswered questions to the shortlisting survey or shorten the list to make the survey quicker to complete. The group chose not to remove any questions.

To ensure questions were easy to understand, they were reviewed by patient and parent members of the steering group and a health information specialist from one of the funding charities. Questions were simplified following this review and definitions of words added.

#### Shortlisting survey
The shortlisting survey was created using Qualtrics software, launched 3 August 2022 and closed 30 September 2022. Invitations mirrored the initial survey, and it was publicised using the same methods. Initial survey participants who left contact details were emailed directly.

To shorten the question list, respondents were invited to read the 101 questions and select those that were most important to them. Questions selected were added to their own personal 'shortlist' ready for them to make their final selection of up to 15 questions. Survey fatigue was minimised by randomisation of section order and questions. This randomisation aimed to limit question selection bias, for example, always selecting the first or last presented questions.

Questions were grouped into:
1. Side effects and management.
2. Treatment.

---

> **Box 1 Out-of-scope question categories and examples**
>
> 1. The question was ambiguous, was interpreted in different ways by steering group members and the meaning could not be resolved following discussion:
>    'Remaining scar tissue'
>    'How research is going'
> 2. Questions not answerable by research:
>    'Why does paediatric cancer research receive so very little funding?'
>    'Who is present when you give the diagnosis'.
> 3. Questions submitted by people whose experience was not of childhood cancer as defined by our project scope—there were a few parent respondents whose child was over 16 at diagnosis. These questions were checked to verify that all the themes within them had been covered by 'in scope' questions.

---

3. Education.
4. Physical activity, play and therapies.
5. Long-term effects and follow-up care.
6. Communication and information sharing.
7. Psychological and social well-being.
8. Food and nutrition.
9. Healthcare delivery.
10. Causes of cancer, diagnosis and research.

Results were analysed in three groups: (1) patients/survivors, (2) parents/friends/relatives and (3) professionals. This gave equal weight to each group's choices as more parents/friends/relatives took part. Questions were given a rank depending on the number of votes and ordered from highest to lowest for each group. The steering group reviewed and compared respondent groups and decided to take the Top 10 questions for each of the three groups to the workshop. This ensured that what was important to each group would be considered and resulted in 21 questions being shortlisted, as some questions were shared priorities.

### Stage 4a: workshop with children and young people
The children and young people's workshop took place in October 2022. The workshop was facilitated by SA and FG following the methodology used by the Juvenile Idiopathic Arthritis PSP.[8] Children were given a choice of seven envelopes, each containing questions on a different topic with a total of 31 questions. Topics were as follows:
1. Family, friends and pets.
2. Treatments and medicines.
3. Being poorly, side effects and long-term effects.
4. Being in hospital.
5. Emotions, worries and getting help or support.
6. School and education.
7. Getting the information you need.

Each participant chose the topic which was most important to them. Envelopes were opened, and participants placed the questions on the table in groups of most, medium or least important. Participants were invited to add more questions if anything of importance to them was missing. They were given three stickers to vote for their top three questions. Questions were placed in order of most to least votes and a discussion followed to agree the 'Top 5'; these were taken to the final workshop.

### Stage 4b: Top 10 prioritisation
The final prioritisation workshop took place in November 2022. Participants who left their contact details in the survey were invited to attend as were patient and parent representatives on the steering group. Steering group contacts were used to ensure participation from a broad range of professionals across the field.

Prior to the workshop, participants were asked to individually rank the questions in order of importance. The workshop was chaired by JG and supported by two JLA facilitators. Participants were split into three preallocated groups ensuring a balance of multidisciplinary professionals, young adults and parents/relatives. In each group, participants shared their three highest and lowest ranking questions. Participants were told which questions were in the children's Top 5.

During facilitated discussion, the groups ordered the questions from highest to lowest priority. The ranking from the three groups were combined. In a second session, groups were reallocated and the combined ranking was discussed. Following this discussion, the group rankings were again collated, and all participants formed one group to debate and agree the Top 10.

### Patient and public involvement
Parent and patient representatives were involved as equal members of the steering group and in all stages of the prioritisation process. Patients and carers were survey respondents. Children were included in a parallel process. Young adults and parents/relatives attended the final prioritisation workshop alongside professionals as equal stakeholders. Participants were reimbursed for travel/overnight accommodation costs.

## RESULTS
Figure 2 provides an overview of the number of respondents at each stage.

### Initial survey
Four-hundred and eighty-eight people submitted 1299 questions. Respondents included 49 (10%) patients/survivors, 291 (60%) parents/relatives/friends and 148 (30%) professionals. Most parents/relative/friends were parents (n=271; 93%), 15 (5%) were relatives and 5 (2%) friends. Online supplemental material 1 shows respondent demographics.

One-hundred and thirty-nine out-of-scope questions were removed; box 1 illustrates examples. Following the combining of similar questions and rewording to form summary questions, 108 questions remained.

### Analysis of uncertainties
Four questions were already answered, and three the focus of ongoing studies. For some questions, no reviews or ongoing studies were identified. If reviews only partly answered a question, these were recorded as unanswered. The steering group discussed all questions ensuring consensus agreement of answered/unanswered questions; 101 questions were unanswered.

### Children and young people's surveys
Seventy-one respondents submitted 252 questions/topics. Sixty-one respondents were children and young people who had experienced cancer (aged 3–21) and 10 were siblings (aged 4–19). No friends participated. See online supplemental material 2 for demographics. For brevity, we refer to submissions as 'questions'; nearly all submissions were not written as questions. Thirteen questions were identified as out-of-scope and removed. Responses were summarised into 24 questions.

**Table 1** Children and young people's Top 5 and questions for the final workshop

| Rank | Top 5 questions from the children and young people's workshop | Question going to the final workshop from the shortlisting survey |
|---|---|---|
| 1 | How can we make being in hospital a better experience for children and young people? (like having better food, internet, toys and open visiting so other family members can be more involved in the child's care) | |
| 2 | How can we prevent cancer in children and young people? | Why do children develop cancer (including the role that genetics plays) and could it be prevented? |
| 3 | How can we make more accessible treatments that are closer to home, in shared care hospitals? | |
| 4 | How can we speed up the process of getting diagnosed and starting treatment in the right place? | How can time to diagnosis be improved for children with suspected cancer? |
| 5 | What are the best ways to help children and young people with their worries and make them feel happier? | What are the best ways to provide emotional support for children and their families (1) around the time of diagnosis, (2) during treatment and (3) after treatment (including survivors who are now adults)? |

## Shortlisting survey

Ratings were submitted by 327 respondents. Like the initial survey, the largest respondent group was parents/relatives/friends (64%, n=210; including 197 parents, 10 relatives, three friends), followed by professionals (28%, n=90) and patients/survivors (8%, n=27). See online supplemental material 3 for demographics.

## Children and young people's workshop

Eight children and young people aged 8–16 attended; three were siblings. Their diagnoses included lymphoma and leukaemia.

During discussion, seven additional questions were created about family, friends and pets and six were added on topics that were important to participants. The Top 5 are shown in table 1. Three of the questions were closely aligned to those already going to the final workshop from the shortlisting survey (priorities 2, 4 and 5). For priority 4, the children and young people's version of the question had an extra part about starting treatment in the right place, this version was taken to the final workshop. Priorities 1 and 3 from children and young people were new and were added into the list, making 23 questions in total for the final workshop.

## Final workshop

Twenty-five participants attended: 4 young adults who had experienced childhood cancer, 5 parents and 1 grandparent of a child who had cancer, and 15 professionals who work with this population. Professional roles varied and included nurses, doctors, a social worker, health play specialist, dietitian, clinical psychologist, physiotherapist and chaplain. One participant was a steering group member.

## Top 10 prioritisation strategies

Although the three groups worked independently, they all applied similar prioritisation strategies:

### Ensuring children's views were represented

All groups wanted to ensure the Top 10 questions included most, if not all, questions from the children's Top 5. When the groups were told which questions were important to children, those question cards were picked out and moved up the ranking. Most of these questions remained in the Top 10, or just outside, for the duration of the discussions.

### Opting for questions that could include other questions/overlap

Groups considered which questions overlapped and could cover other questions. For example, 'can we find effective and kinder (less burdensome, more tolerable, with fewer short and long-term effects) treatments for children with cancer, including relapsed cancer?' mentions side effects and so could include, 'what are the best ways to reduce, predict and manage the side effects of treatment for children (including life-threatening side effects)?

### Opting for questions focused on intervention rather than description

Groups were clear that although it is useful to describe a problem, it is action through intervention that is required to improve children's and families' experiences. Therefore, 'are the psychological, practical and financial support needs of children with cancer, survivors and their families being met during treatment and beyond? How can access to this support be improved and what further support would they like?' was placed higher in the rankings than 'what is the psychological and social impact of cancer and treatment on children and their families during treatment and in the long term; what factors affect these impacts?' as the latter question involves description, rather than action.

### Opting for questions that could have wider impact

Initially, most participants selected their top three questions reflective of their personal experience or area

**Box 2** Top 10 research priorities for children's cancer and the additional 13 questions discussed at the workshop

1. Can we find effective and kinder (less burdensome, more tolerable, with fewer short and long-term effects) treatments for children with cancer, including relapsed cancer?
2. Why do children develop cancer (including the role that genetics plays) and could it be prevented?*
3. Are the psychological, practical and financial support needs of children with cancer, survivors and their families being met during treatment and beyond? How can access to this support be improved and what further support would they like?†
4. How can we speed up the process of getting diagnosed and starting treatment in the right place?*
5. Why do children relapse, how can it be prevented and what are the best ways to identify relapse earlier?
6. How can we make being in hospital a better experience for children and young people? (like having better food, internet, toys and open visiting so other family members can be more involved in the child's care)*
7. What are the best ways to ensure children and families get and understand the information they need, in order to make informed decisions, around the time of diagnosis, during treatment, at the end of treatment and after treatment has finished?
8. What impact does cancer and treatment have on the lives of children and families after treatment, and in the long term; what are the best ways to help them to overcome these impacts to thrive and not just survive?
9. How can we make more accessible treatments that are closer to home, in shared care hospitals?*
10. What is the relationship between chronic fatigue syndrome, fibromyalgia, chronic pain and treatment for childhood cancer? (Fibromyalgia is a long-term condition that causes pain all over the body.)
11. What are the best ways to provide emotional support for children and their families (1) around the time of diagnosis, (2) during treatment and (3) after treatment (including survivors who are now adults)?*
12. What are the best ways to reduce, predict and manage the side effects of treatment for children (including life-threatening side effects)?
13. How can transition (moving) from child into adult services be improved for young people who had cancer as a child?
14. What is the psychological and social impact of cancer and treatment on children and their families during treatment and in the long term; what factors affect these impacts?
15. How common are the different long-term effects of childhood cancer treatment, how do they change across the lifespan, can we predict them and how can they best be prevented, detected and/or treated?
16. What are the best ways to support the emotional well-being of professionals who care for children with cancer and their families?
17. During and after treatment, what issues prevent or encourage physical activity, which interventions are most effective and what should be measured to assess effectiveness?
18. What are the best ways of making sure people who had cancer as a child receive the information they need about the long-term effects of cancer and treatment?
19. What fertility preservation options work best for children and teenagers with cancer?

Continued

**Box 2** Continued

20. What are the long-term effects of additional medications children with cancer may receive (such as antibiotics, pain killers, laxatives) and how can these effects be reduced?
21. What are children's and survivors' experiences of the side effects and long-term effects of cancer treatment?
22. How can experiences of having a Hickman line be improved for children with cancer? (A Hickman line is a small tube which is inserted into a vein so that treatments can be given, and blood taken without the repeated need to access veins with a needle. The Hickman line can stay in place for several months.)
23. What are the best ways to support children as they get older, and their needs change, to understand and take responsibility for their health, and to live with the long-term effects of cancer and treatment?

*These questions were in the Top 5 research priorities identified by children and young people.
†This question was originally not mapped onto the question about emotional support from children and young people, but the workshop participants decided that this question was related as it includes emotional support as well as other types of support.

they worked within. During discussions, their opinions changed, and groups decided that the Top 10 questions should be generic and have the potential to have the greatest impact on as many children and families as possible. For example, 'how can experiences of having a Hickman line be improved for children with cancer?' was considered too specific and did not apply to all children.

### Ensuring all themes within the questions were represented

Groups tried to cluster questions into similar themes, such as support, treatment, care, side effects, their aim being to include each 'theme' in the Top 10. For example, the question about relapse was moved up during discussions as this was not covered by any other question.

### Group discussion and decision-making

From the outset, there were some questions that were high priority for many and stayed high in the Top 10 throughout the workshop. The question ranked as top priority, 'can we find effective and kinder (less burdensome, more tolerable, with fewer short and long-term effects) treatments for children with cancer, including relapsed cancer?' was the top priority for all three groups after the first group discussion. After the second group discussion, all three groups had the same questions ranked one to five, which remained in the same positions in the final Top 10.

The final group discussion focused on whether to include, 'what is the relationship between chronic fatigue syndrome, fibromyalgia, chronic pain and treatment for childhood cancer?' in the Top 10 (it was at number 11). This push for inclusion came from two young adults who said these long-term effects had a huge impact on their lives and had experienced a lack of recognition and support. There was a group vote and the decision was

made to move this question up to number 10 and move, 'what are the best ways to provide emotional support for children and their families (1) around the time of diagnosis, (2) during treatment and (3) after treatment (including survivors who are now adults)?' down to number 11 as this was covered by the broader question, about support at number 3.

The final Top 10 priorities are shown in box 2 alongside the other 13 questions discussed.

## DISCUSSION

The Children's Cancer PSP brought together children, survivors, families and professionals to prioritise research questions on childhood cancer. The Top 10 priorities provide a resource to inform research funding decisions in government and charitable organisations. The top priority is 'can we find effective and kinder (less burdensome, more tolerable, with fewer short and long-term effects) treatments for children with cancer, including relapsed cancer?' This question was ranked as top in the shortlisting survey by all three respondent groups (patients/survivors, parents/relatives/friends and professionals) and placed at number 1 from the start of the workshop by all three discussion groups. This reflects shared priorities of continuing to improve cure rates while minimising treatment toxicity. The Top 10 priorities reflect the breadth of the cancer experience, including diagnosis, relapse, hospital experience, support during/ after treatment and the long-term impact. Priorities highlight the need for research strategies to be holistic in their approach rather than solely driven by biological and drug intervention research. It is now critical that funders and researchers ensure future research focuses on what is important to children, survivors, families and professionals.[9]

A number of cancer-related PSPs exist, including one in Canada also focusing on childhood cancer (https:// www.jla.nihr.ac.uk/priority-setting-partnerships/pediatric-cancer/top-10-priorities.htm). The top priority for the Canadian PSP is preventing and managing treatment-related long-term effects which links to the top priority of our PSP and finding 'kinder' treatments. Both Top 10 lists feature similar questions on relapse, prevention/ detection and questions about psychosocial impact and support. There is an increasing drive to focus on both physical and psychological health during and after cancer. It is already recognised that a cancer diagnosis has serious implications for children and young people's mental health during and after treatment,[10 11] but this has yet to be systematically investigated, and how best to provide support remains unknown. Psychological support was the top priority in the Teenage and Young Adult Cancer PSP.[12]

### Challenges, strengths and limitations

The anticipated timeline for this project was two years, it took three. This delay was partly due to the COVID-19 pandemic. The project was resource intensive, requiring input from all steering group members. The challenge of involving professionals with full schedules, and parent/ patient representatives with many concerns and commitments, was amplified by the pandemic and our progress reflected this.

The scope of the PSP was intentionally broad to reflect the heterogeneity of childhood cancer, and variation in treatment and experience. This generated a significant workload when sorting and summarising diverse questions, and subsequent literature searching to verify uncertainties.

Engaging with children extended the project timescale; this work had to be carefully planned to ensure our methods were accessible and appropriate. Plans for face-to-face work were revised due to pandemic restrictions. Few priority setting exercises have involved many children and young people.[6 13] Previous PSPs have reflected that they were unable to engage with children as they wished, due to lack of time and resources.[14] It was of utmost importance to our steering group that children's voices were heard. We consider this aspect of our PSP a success: time and resources invested in engaging with children were worthwhile. Overall, questions from children reflected similar themes as those from adult participants, but there were some additional elements that featured as higher priority for children, such as having treatments closer to home and improving the hospital experience. In the final workshop, participants wanted children's voices to be heard, resulting in all five of the top priorities identified by children being reflected in the Top 10.

The use of the rigorous and transparent JLA methodology enhances the validity of the process and results. The response from parents/carers to both surveys was high and parent and patient representatives were involved in shaping the project from the outset, as members of the steering group. Their input was key, for example, they helped to ensure the surveys were presented in a user-friendly format and appropriate routes to dissemination were used. Parent/patient representatives reported a positive experience of being involved in the steering group, *'I wanted to be involved with the PSP because of the exciting opportunity to contribute towards future research topics in childhood cancer, bringing the voice of childhood cancer survivors from a service user perspective and advocating for the cohort. I have found the experience to be extremely positive and engaging. I feel that my presence is valued, and my contributions have been acknowledged and implemented throughout the process.'*

Absent voices must be considered as a limitation. Of note, the majority of respondents described themselves as White. The priorities, therefore, represent the views of the majority, White population, which has been observed in other PSPs.[15] Males were also under-represented. We did not ask in the surveys whether respondents have a disability (whether resulting from treatment or not) and so cannot comment on what impact this might have had on prioritisation.

Primary care has an important role in the care of children with cancer from diagnosis into survivorship.[16] There was a primary care representative on the steering group and at the final workshop, but none responded to the initial survey, and only one to the shortlisting survey. The voices of these professionals are absent from the questions collected.

## IMPLICATIONS AND DISSEMINATION

The Top 10 have been circulated on social media and via supporter newsletters/websites by the PSP funding charities and our Partners. Dissemination includes publication of a final report with an associated launch event, peer-reviewed publications and conference presentations. We will report the details of our engagement with children in a separate publication and are working with the JLA to develop guidance for future PSPs.

Our aspiration is that these prioritised questions will help to direct and shape future research. The uncertainties identified are the outcome of a systematic and transparent process and provide funders with clear guidance on the highest priorities for future research, voted on by end-users of research. Identifying clear areas for future research allows research funders to target funds effectively and inform fundraising activities. We plan to hold a meeting with funders to promote the priorities and encourage funding calls focused on the priority areas.

When selecting questions to be included in the Top 10, workshop participants intentionally opted for broad questions, to capture the widest range of issues. This is common in JLA PSPs, the questions, therefore, reflect broad topic areas for research; further refinement is required to transform topics into answerable research questions.[17] This PSP also demonstrates that where sufficient expertise and resources are available, involvement of young children can be achieved. Therefore, funding guidance should encourage applicants to undertake such work.

Some questions submitted were outside the scope of the PSP and were removed. Many suggested a knowledge gap. The steering group considered these questions to be important and is determined to ensure these submissions are not 'lost'. We will look at how these questions, statements and service enquiries can be best used to improve information signposting. Questions were submitted regarding disparity in funding between childhood and 'adult' cancers. These questions were removed, as they are not amenable to research, but we intend to share them through a commentary piece, as they reflected strong opinions and perceptions that would benefit from further exploration and articulation.

## CONCLUSION

We have identified shared research priorities for children's cancer using a rigorous, person-centred approach involving stakeholders not typically involved in setting the research agenda, including children. Resulting questions reflect the breadth of the cancer experience for children and families, including diagnosis, relapse, hospital experience, support during and after treatment and the long-term impact of cancer. These must inform funding of future research, with priority questions evidenced by researchers.

**Author affiliations**
[1]School of Health Sciences, University of Surrey, Guildford, UK
[2]Leeds Children's Hospital, Leeds, UK
[3]Department of Paediatric Haematology and Oncology, Leeds Teaching Hospitals NHS Trust, Leeds, UK
[4]Hull-York Medical School and Centre for Reviews and Dissemination, University of York, York, UK
[5]Children's Cancer and Leukaemia Group, Leicester, UK
[6]Patient Representative on the Children's Cancer Priority Setting Partnership Steering Group, London, UK
[7]The Royal Marsden NHS Foundation Trust, Sutton, UK
[8]Institute of Cancer Research Sutton, Sutton, UK
[9]Parent Representative on the Children's Cancer Priority Setting Partnership Steering Group, Coventry, UK
[10]Bristol Royal Hospital for Children, Bristol, UK
[11]James Lind Alliance, National Institute for Health Research Evaluation, Trials and Studies Coordinating Centre, Southampton, UK
[12]Southampton Children's Hospital, Southampton, UK
[13]University of Southampton Faculty of Medicine, Southampton, UK
[14]Alder Hey Children's NHS Foundation Trust, Liverpool, UK
[15]University Hospitals Birmingham NHS Foundation Trust, Birmingham, UK
[16]Parent Representative on the Children's Cancer Priority Setting Partnership Steering Group, London, UK
[17]Young Lives Vs Cancer, London, UK
[18]Medical Needs Teaching Service, Leeds Children's Hospital, Leeds, UK
[19]Centre for Reviews and Dissemination, University of York, York, UK
[20]Children, Teenage and Young Adult Cancer Operational Delivery Network, South West, Bristol, UK
[21]Royal Devon and Exeter NHS Foundation Trust, Exeter, UK
[22]Psychological Services, Great Ormond Street Hospital for Children NHS Foundation Trust, London, UK
[23]The Christie NHS Foundation Trust, Manchester, UK
[24]Parent Representative on the Children's Cancer Priority Setting Partnership Steering Group, Perth, UK
[25]The Little Princess Trust, Hereford, UK
[26]Parent Representative on the Children's Cancer Priority Setting Partnership Steering Group, Keswick, UK
[27]The University of Edinburgh, Edinburgh Medical School, Edinburgh, UK
[28]Centre for Outcomes and Experience Research in Child Health, Illness and Disability (ORCHID), Great Ormond Street Hospital For Children NHS Foundation Trust, London, UK

**Acknowledgements** The Children's Cancer PSP would like to thank everyone who took the time to send in their questions and vote on the importance of them. Thank you also to the children, young people, parents, relatives and professionals who attended the workshops. We would like to thank Angela Stewart for providing administrative support to the PSP. We would also like to thank the previous members of the steering group: Martin English, Penelope Hart-Spencer, Charmaine Jagger and Angela Polanco.

**Contributors** All authors (SA, RH, BP, AB-G, AB, JC, SC, RD, JG, NJH, HH, JH, LH, LL, KiM, SM, KeM, JEM, HM, SPa, SPi, RR-B, DS, AS, WT-M, AWalsh, AWatkins, DW and FG) were part of the Children's Cancer Priority Setting Partnership steering group or coordinating team and made substantive contributions to the conduct of the study, overseeing all aspects of the work. All authors contributed to protocol design, survey refining, data cleaning and refining questions submitted in the initial survey. The project was managed by SA and FG (guarantor), BP, RH and JG. SA, FG, JEM, SM, LL, KeM and RR-B were part of a subgroup overseeing engagement

with children throughout the PSP process. Specific contributions included: survey design (SA), coding the survey submissions (FG and SA), searching and checking uncertainties (SA, BP), managing data entry (SA). All authors reviewed and approved the final version of this paper.

**Funding** This work was supported by Children's Cancer and Leukaemia Group (CCLG) and Little Princess Trust. No grant award number available. Dr Julia Chisholm is supported by the Giant Pledge through the Royal Marsden Cancer Charity and this independent research is supported by the National Institute for Health Research (NIHR) Biomedical Research Centre at The Royal Marsden NHS Foundation Trust and the Institute of Cancer Research, London. The views expressed are those of the authors and not necessarily those of the NIHR or the Department of Health and Social Care. FG is supported in-part by the Great Ormond Street NIHR Biomedical Research Centre.

**Competing interests** None declared.

**Patient and public involvement** Patients and/or the public were involved in the design, or conduct, or reporting, or dissemination plans of this research. Refer to the Methods section for further details.

**Patient consent for publication** Not applicable.

**Ethics approval** This study involves human participants but ethical approvals are not required for JLA Priority Setting Partnerships as per JLA and National Health Services Patient Safety Agency National Research Ethics Service guidance (https://www.invo.org.uk/posttypepublication/public-involvement-in-research-and-research-ethics-committee-review/). The James Lind Alliance does not define priority setting exercises as a research study.

**Provenance and peer review** Not commissioned; externally peer reviewed.

**Data availability statement** All data relevant to the study are included in the article or uploaded as online supplemental information. Further data regarding the original submissions to the surveys are available from: https://www.jla.nihr.ac.uk/priority-setting-partnerships/childrens-cancer/.

**ORCID iDs**
Susie Aldiss http://orcid.org/0000-0002-6015-6993
Ashley Ball-Gamble http://orcid.org/0000-0002-0708-0918
Nigel J Hall http://orcid.org/0000-0001-8570-9374
Jessica Elizabeth Morgan http://orcid.org/0000-0001-8087-8638
Faith Gibson http://orcid.org/0000-0002-8125-4584

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
