## [Reviewer comments · BMJ Open]

ARTICLE DETAILS

TITLE (PROVISIONAL)	Research priorities for children's cancer: a James Lind Alliance priority setting partnership in the United Kingdom
AUTHORS	Aldiss, Susie; Hollis, Rachel; Phillips, Bob; Ball-Gamble, Ashley; Brownsdon, Alex; Chisholm, Julia; Crowther, Scott; Dommatt, Rachel; Gower, Jonathan; Hall, Nigel; Hartley, Helen; Hatton, Jenni; Henry, Louise; Langton, Loveday; Maddock, Kirsty; Malik, Sonia; McEvoy, Keeley; Morgan, Jessica; Morris, Helen; Parke, Simon; Picton, Sue; Reed-Berendt, Rosa; Saunders, Dan; Stewart, Andy; Tarplee-Morris, Wendy; Walsh, Amy; Watkins, Anna; Weller, David; Gibson, Faith

VERSION 1 – REVIEW

REVIEWER	Jibb, Lindsay Hospital for Sick Children
REVIEW RETURNED	31-Jul-2023

GENERAL COMMENTS	My sincere thanks for the opportunity to read through the paper entitled: "Research priorities for children's cancer: a James Lind Alliance priority setting partnership in the United Kingdom". This paper provides essential direction to the next research to be undertaken in the area of children's cancer by providing insights from those impacted most by the disease. A specific strength of this paper is the time, effort, and attention put into engaging with younger children with cancer—a group hardly reached in direct research (but, again, most often most impacted). As a researcher in the field, I am grateful to the authors for this contribution to the literature. I have a few comments/thoughts for consideration – but these are very minor. The authors could consider adding in the numbers of children and young people who piloted the children and young people's survey for consistency with reporting for initial survey. On Page 7 lines 10-12, would the authors consider highlighting if siblings and friends had to have a brother, sister or friend who is before their 16th birthday? On Page 8, line 57, the authors mention that "survey fatigue was minimized by randomization of section order and questions". I wonder if the authors might consider highlighting here the strength of randomization in potentially limiting question section bias (e.g., bias towards always selected first presented or last presented questions). Related to the above point, it might be nice to present this line prior to the list of question grouping (keeping it together with text related to the shortlisting process).
--

	Supplementary material 2: I did not see the ages of children who responded to the children and young people's survey presented. If these data are available to the authors, I would recommend including in the table as I think a great strength of this paper is the engagement of children from younger age groups and typically associated developmental stages. Once again thank you for this paper.
--	--

REVIEWER	Rowbotham, Nicola University of Nottingham School of Medicine, Division of Child Health, Obstetrics and Gynecology
REVIEW RETURNED	15-Oct-2023

GENERAL COMMENTS	Thank you for inviting me to review this well written and thought out manuscript. The authors' aim is to systematically identify and prioritise research questions of importance to the childhood cancer community. The authors do this through a James Lind Alliance Priority Setting Partnership which is gold standard methodology for this type of work. The authors amended their study methods slightly from traditional JLA methodology, with a parallel process for the under 16s to ensure that the voices of children were adequately heard and included in the PSP. Through this, a series of surveys and a workshop a final top ten list of priorities for research in children's cancer were agreed. I feel it is important to publish this paper as it will help disseminate the top ten questions, increase awareness to funders and focus future research in childhood cancer to those questions of most importance to those living through and caring for those with childhood cancer. It will also be of interest to others engaging in PPI work and JLA PSPs as the extra methodologies to ensure the voice of the child is included will be useful for other disease areas. The manuscript is incredibly well written, thorough and interesting to read. All the extra information not included that one might need, can be found through the links to the JLA website. I cannot find any suggestions for improvement so recommend that it should be accepted without needing revision.
--

VERSION 1 – AUTHOR RESPONSE

Reviewer comment	Changes made
The authors could consider adding in the numbers of children and young people who piloted the children and young people's survey for consistency with reporting for initial survey.	The number of children who piloted the surveys has been added.
On Page 7 lines 10-12, would the authors consider highlighting if siblings and friends had to have a brother, sister or friend who is before their 16th birthday?	We have added this information into the manuscript.

On Page 8, line 57, the authors mention that “survey fatigue was minimized by randomization of section order and questions”. I wonder if the authors might consider highlighting here the strength of randomization in potentially limiting question section bias (e.g., bias towards always selected first presented or last presented questions). Related to the above point, it might be nice to present this line prior to the list of question grouping (keeping it together with text related to the shortlisting process).	We have expanded on why we used randomisation by adding the sentence, ‘This randomisation aimed to limit question selection bias, for example always selecting the first or last presented questions.’ We have also moved this information up to the location the reviewer suggested.
Supplementary material 2: I did not see the ages of children who responded to the children and young people’s survey presented. If these data are available to the authors, I would recommend including in the table as I think a great strength of this paper is the engagement of children from younger age groups and typically associated developmental stages.	We agree that this information should be included and it has now been added to supplementary material 2.